# Non-Coding RNAs: lncRNA, piRNA, and snoRNA as Robust Plasma Biomarkers of Alzheimer’s Disease

**DOI:** 10.3390/biom15060806

**Published:** 2025-06-03

**Authors:** Ruomin Xin, Elizabeth Kim, Wei Tse Li, Jessica Wang-Rodriguez, Weg M. Ongkeko

**Affiliations:** 1Department of Otolaryngology—Head and Neck Surgery, University of California, La Jolla, San Diego, CA 92093, USA; rxin@ucsd.edu (R.X.); harrison.li@ucsf.edu (W.T.L.); 2Research Service, VA San Diego Healthcare System, San Diego, CA 92161, USA; 3School of Medicine, University of California, San Francisco, CA 94143, USA; 4Department of Pathology, University of California, La Jolla, San Diego, CA 92093, USA; jwrodriguez@ucsd.edu; 5Pathology Service, VA San Diego Healthcare System, San Diego, CA 92161, USA

**Keywords:** Alzheimer’s disease, non-coding RNA, disease diagnostics, transcriptomics, gene regulation, liquid biopsy, biomarkers

## Abstract

Alzheimer’s disease (AD) is a leading cause of dementia worldwide. As current diagnostic approaches remain limited in sensitivity and accessibility, there is a critical need for novel, non-invasive biomarkers aiding early detection. Non-coding RNAs (ncRNAs), including long non-coding RNAs (lncRNAs), PIWI-interacting RNAs (piRNAs), and small nucleolar RNAs (snoRNAs), have emerged as promising candidates due to their regulatory roles in gene expression and association with diseases. In this study, we systematically profiled ncRNA expression from RNA sequencing data of 48 AD and 22 control blood tissue samples, aiming to evaluate their utility as biomarkers for AD classification. Differential expression analysis revealed widespread dysregulation of lncRNAs and piRNAs, with over 5000 lncRNAs and nearly 1000 piRNAs significantly upregulated in AD. Weighted gene co-expression network analysis (WGCNA) identified multiple ncRNA modules associated with the AD phenotype. Using supervised machine learning approaches, we evaluated the diagnostic potential of ncRNA expression profiles, including single-gene, multi-gene, and module-level models. Random Forest models trained on individual genes identified 121 ncRNAs with AUROC > 0.8. Feature importance analysis emphasized ncRNAs such as lnc-MYEF2-3, lnc-PRKACB2, and HBII-115 as major contributors to diagnostic accuracy. These findings support the potential of ncRNA signatures as reliable and non-invasive biomarkers for AD diagnosis.

## 1. Introduction

Alzheimer’s disease (AD) is the seventh leading cause of death, affecting over 6 million individuals in the United States alone [1]. Influenced by multiple risk factors such as age, genetics, lifestyle, cardiovascular health, and history of head trauma, the current understanding of AD’s complex pathogenesis is still limited, posing significant challenges to the development of effective AD treatment [2]. The existing U.S. Food and Drug Administration (FDA) approved prescription drugs for AD work towards alleviating the symptoms of AD rather than curing the disease [2]. Popular drugs, including cholinesterase inhibitors, work best only for people in the earlier stages of AD, thus highlighting the critical need for early AD diagnosis to enable effective preventive care [3].

Current methods of diagnosis utilize medical history, physical and mental status examination, neuroimaging, and cerebrospinal fluid (CSF) biomarker tests [3]. The more accurate methods of neuroimaging and CSF tests are often only administered after the initial assessments using behavioral tests. However, behavior observations are prone to misinterpretations and demonstrate underestimation of symptoms when patients become familiar with specific diagnosis tasks, prompting the search for a reliable and non-invasive early diagnostic tool [4]. Recently, Barthélemy et al. constructed a plasma %p-tau217 test that had an area under the curve (AUC) value between 0.95 and 0.98, which was equivalent to the predictive strength of FDA-approved CSF tests (AUC = 0.95–0.97) [5]. While further research is still required to validate this finding, the non-invasive nature of blood-based tests and promising capabilities in AD detection place this method at the center of an investigation to unravel more reliable biomarkers.

Past investigation efforts in AD pathology have focused on amyloid plaques and the development of neurofibrillary tangles. A previously understudied class of biomolecules, non-coding RNAs (ncRNAs), is increasingly being studied for their role in AD progression. Small nucleolar RNAs (snoRNAs), a type of ncRNA, participate in rRNA modification [6]. Studies have linked the dysregulation of snoRNAs to cognitive decline in AD due to their regulatory role in protein synthesis and RNA processing [7,8]. Differential expressions of snoRNAs were observed in plasma extracellular vesicles, providing an accessible type of diagnostic biomarker [7]. Furthermore, in mouse models, dysregulation of snoRNAs was identified before the formation of amyloid plaques, suggesting the promising potential of snoRNA in early diagnosis of AD [9].

PIWI-interacting RNAs (piRNAs), another type of ncRNA, play a role in transposon silencing and may exhibit transgenerational inheritance [10]. Studies reveal more than 103 differentially expressed piRNAs in AD brain tissue and also showed that most of these piRNAs correlated with AD risk SNPs and AD-associated signaling pathways [11,12]. Additionally, genome-wide association studies have revealed a correlation between risk variants and piRNA expression [13]. A combined signature of microRNAs and piRNAs offers a promising strategy for early diagnosis of AD, yet larger and more diverse cohorts are needed to validate the findings [14].

Long non-coding RNAs (lncRNAs), transcripts longer than 200 nucleotides, are another class of ncRNAs that regulate translation, metabolism, and cellular signaling [15]. Studies have shown that dysregulation of specific lncRNAs in AD may serve as diagnostic biomarkers [13,16]. In a study identifying 11 differentially expressed lncRNAs in AD blood samples, the RN7SK LncRNA had a sensitivity and specificity of 50% and 84%, respectively (AUC = 0.658, *p* = 0.0063) [16]. In AD brain samples, BIN1 has also been shown to transcriptionally modulate mRNA expression and influence the progression of AD [13]. In addition, levels of lncRNA BACE1 in plasma were significantly elevated with AD, showing an AUC of 0.667 [17]. Overall, existing studies underscore the role of snoRNAs, piRNAs, and lncRNAs as novel biomarkers for the early diagnosis of AD.

Hence, this study aims to evaluate the potential of ncRNA expression profiles as predictive biomarkers for AD. RNA sequencing data of AD and control blood samples were downloaded from the NCBI Sequence Read Archive (SRA). The dataset was analyzed to quantify expression levels of lncRNA, piRNA, and snoRNA. Weighted gene co-expression network analysis revealed co-expression modules correlating strongly with AD. Furthermore, ncRNA expression profiles were used to train supervised machine-learning algorithms. A nested cross-validation approach was used to benchmark model performances. We evaluated the diagnostic potential of ncRNA expression profiles, including single-gene, multi-gene, and module-level models. Through this investigation, we provide compelling evidence that ncRNA signatures can serve as reliable biomarkers for predicting AD, potentially offering novel insights into disease progression and therapeutic targets of AD.

## 2. Materials and Methods

### 2.1. Data Acquisition

RNA sequencing data were obtained from a publicly available dataset from the NCBI SRA database (https://www.ncbi.nlm.nih.gov/sra) under project accession SRP022043 (https://www.ncbi.nlm.nih.gov/Traces/study/?acc=SRP022043&o=acc_s%3Aa) (accessed on 3 March 2025) and SRP288246 (https://www.ncbi.nlm.nih.gov/bioproject/PRJNA670793) (accessed on 20 May 2025) [18,19]. The main study includes 70 individuals, comprising 48 Alzheimer’s disease patients and 22 healthy controls. Plasma samples were sequenced using the Illumina HiSeq 2000 platform (San Diego, CA, USA) in a single-end format. An additional 50 samples, including 25 diseased patients and 22 healthy controls, were processed similarly from the secondary dataset. Postmortem brain samples were sequenced using the Illumina HiSeq 4000 platform in single-end format. FASTQ files were downloaded and extracted using SRA Toolkit v3.2.0 [20].

### 2.2. Quality Control and Preprocessing

Quality control was performed using FastQC v0.12.1 to assess sequencing read quality [21]. Samples were assessed based on per-base sequence quality, per-sequence quality, sequencing content, overrepresented sequences, and adapter content. Samples SRR837499, SRR837500, and SRR837504 were discarded due to low average per-sequence Phred quality scores. The remainder samples were trimmed using cutadapt v1.18 to remove adapter sequences and to filter reads using a Phred score threshold of 20 and a read length filter of 18 base pairs [22].

### 2.3. Alignment to Reference Genome and Feature Annotation

Trimmed reads were aligned to the human reference genome (hg38) using STAR v2.7.2b [23]. The resulting BAM files were sorted by genomic coordinates and mapped to the annotated ncRNA feature from the DASHR v2.0 annotation database for lncRNAs, piRNAs, and snoRNAs [24]. Bedtools v2.26.0 was used to calculate coverage per feature, resulting in raw read counts for downstream analysis [25].

### 2.4. Differential Expression Analysis and WGCNA

Differential expression analysis was performed using DESeq2 v1.46.0 in R [26]. Normalization and dispersion estimation were performed, and differentially expressed genes (DEGs) in AD samples were identified with a threshold of log2 fold change ≥ 0.6 and adjusted *p*-value ≤ 0.05. Weighted gene co-expression network analysis (WGCNA) was performed using WGCNA 1.73 to identify ncRNA co-expression modules associated with Alzheimer’s pathology [27]. Module detection and eigengene computation were conducted using the blockwise modules function with a soft-threshold power of 12, a minimum module size of 30, and hierarchical clustering with dynamic tree cutting. Hypergeometric testing was performed to evaluate the statistical significance of gene overlap between the discovery and validation datasets based on the total number of genes tested and the number of differentially expressed genes in each dataset.

### 2.5. Machine-Learning Models and Feature Importance

To classify Alzheimer’s disease samples based on gene expression profiles, we trained supervised machine-learning Random Forest (RF) models using the Python module scikit-learn 1.6.1 [28]. Models were trained using 70% of the data and evaluated on the remaining 30% using stratified fold cross-validation.

### 2.6. Pairwise Correlation Analysis and Heatmap Generation

Pairwise Spearman correlation analysis was performed on normalized expression values of top predictive ncRNAs to assess co-expression patterns. Using R (with tidyverse v2.0.0, pheatmap v1.0.12), correlation coefficients and *p*-values were calculated for all gene pairs. Significant correlations (*p* ≤ 0.05) were annotated using asterisks [29,30].

## 3. Results

RNA-seq data from 70 samples (48 AD, 22 controls) were obtained. After quality assessment with FastQC and removal of three low-quality samples (SRR837499, SRR837500, SRR837504), reads were trimmed using cutadapt and aligned to the human genome (hg38) using STAR. Mapped reads were annotated using DASHR v2.0 to quantify ncRNA species (lncRNAs, piRNAs, snoRNAs) with Bedtools. Differential expression analysis was performed with DESeq2, followed by weighted gene co-expression network analysis (WGCNA) to identify modules associated with Alzheimer’s disease. Machine-learning models were trained to classify AD vs. control samples and identify key diagnostic features. Model performance was validated using an independent external dataset comprising 50 samples (25 AD, 25 controls) (Figure 1).

**Figure 1 biomolecules-15-00806-f001:**
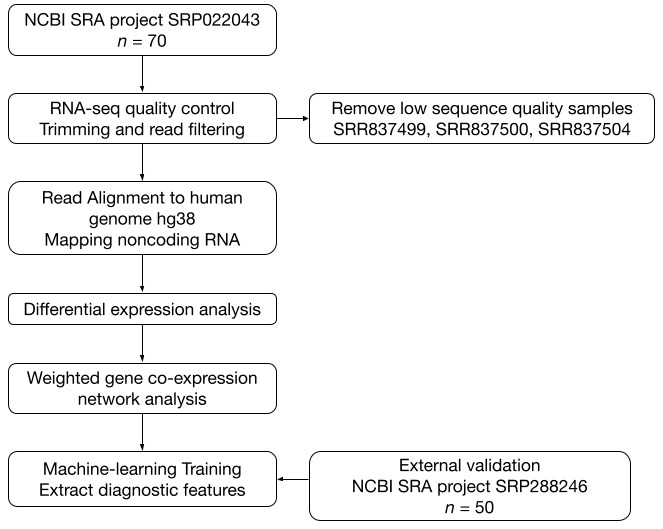
Workflow of the study. Flowchart demonstrates overall methodology employed in this study.

### 3.1. Differential Expression Analysis of ncRNA

Using DESeq2 (adjusted *p*-value < 0.05), we revealed substantial dysregulation of ncRNA transcripts across 48 AD and 22 control blood samples. A total of 5164 lncRNA transcripts were significantly upregulated, while 290 lncRNA transcripts were significantly downregulated in AD samples (Figure 2A,D). The most strongly upregulated lncRNA—lnc-ARL4C-09—displayed approximately a 250-fold increase (log₂ fold change = 7.98), whereas the most strongly downregulated lncRNA—lnc-CTNNBL1-3—displayed approximately a 260-fold reduction (log₂ fold change = *−*8.05) in AD. A total of 973 piRNAs were significantly upregulated in AD, while 26 piRNAs were downregulated (Figure 2B,D). Aside from the smaller set of differentially expressed genes, the most significantly dysregulated piRNAs also demonstrate changes in lower magnitudes. The most significantly upregulated piRNA—piR-47795—is enriched approximately 31-fold (log₂ fold change = 4.97) in AD. Similarly, the most significantly downregulated piRNA—piR-48801—was reduced by around 6-fold in AD (log₂FC = −2.48). The fewest number of differentially expressed genes was observed in the class of snoRNA. Analysis revealed six significantly upregulated and one significantly downregulated snoRNA in AD. The most notable changes were observed for HBII-115 (log₂FC = 1.67) and HBII-55 (log₂FC = *−*1.29), indicating modest yet consistent expression shifts in this class. These findings indicate widespread ncRNA dysregulation in AD, with lncRNAs and piRNAs showing dominant upregulation patterns and a broader range of fold changes compared to snoRNAs.

### 3.2. Weighted Gene Co-Expression Network Analysis

To investigate the modular organization of non-coding RNA expression in Alzheimer’s disease, we performed weighted gene co-expression network analysis (WGCNA) using variance-stabilized expression values. Samples were first hierarchically clustered using pairwise Euclidean distances to identify intra-dataset heterogeneity (Figure 3A). A static cut height was set visually at 85 to remove sample SRR837506 as an outlier before network construction.

Following hierarchical clustering and dynamic tree cutting (Figure 3B), we identified 15 co-expression modules represented by distinct colors. The largest modules included the turquoise module (*n* = 2261 genes) and the blue module (*n* = 2046 genes), followed by the brown (*n* = 783), yellow (*n* = 425), and green (*n* = 155) modules (Table 1). Following module identification, a module–trait relationship heatmap across selected major modules was performed to assess the association of modules with distinct cohorts (Figure 3C). Expression levels of single genes within each module were plotted across samples to further demonstrate the consistency of expression profiles in AD relative to control. These findings highlight the presence of coordinated ncRNA expression modules that are associated with AD’s pathology.

### 3.3. Supervised Machine-Learning Prediction of Alzheimer’s Disease Using ncRNA Expression Profiles

The diagnostic performances of 6459 differentially expressed genes were assessed using a Random Forest (RF) classifier (Figure 4A). Models were trained using single gene expression profiles and evaluated using 5-fold cross-validation to assess the standalone predictive strength of ncRNA. Results revealed substantial variation in the predictive power of individual ncRNA, with an average area under the ROC curve (AUC) of 0.620 ± 0.099 and 121 differentially expressed genes exceeding an AUC of 0.8. The top five performing candidate genes of each ncRNA class were used to construct multi-gene expression panels for the comprehensive evaluation of the classification performance of different ncRNA subtypes. We performed 10-fold, 100-repeat nested cross-validation using a Random Forest classifier. Across all iterations, the lncRNA panel yielded the highest average AUROC (Mean = 0.974, Median = 1.000), followed closely by the combined ncRNA panel (Mean = 0.972), indicating strong discriminative capability (Figure 4B,D). The piRNA panel also achieved high performance (Mean AUROC = 0.925), while the snoRNA panel showed more variability and overall lower predictive strength (Mean AUROC = 0.802). The combined model notably demonstrated high precision (Mean = 0.960) and sensitivity (Mean = 0.918). Feature importance ranking indicates that snoRNAs have higher importance in the combined ncRNA panel prediction, despite weaker standalone predictive strength and snoRNA panel predictive strength (Figure 4C).

Pairwise correlation analysis was conducted to investigate the relationships among the top predictive ncRNAs. Hierarchical clustering revealed two primary groups of ncRNAs with distinct co-expression patterns (Figure 5). One cluster, consisting of lnc-MYEF2-3, lnc-SI-3, and lnc-TSPYL6-1, exhibited strong positive correlations, suggesting coordinated expression. In contrast, another cluster comprising lnc-PRKACB-2, piR-59350, and piR-32371 showed weaker and more variable correlations, indicative of divergent expression profiles. An intermediate group of ncRNAs also formed a positively correlated cluster, suggesting partial co-expression behavior. These clustering patterns highlight the presence of diverse regulatory dynamics among ncRNAs in Alzheimer’s disease.

An alternative Random Forest classifier was trained using the averaged expression profile of 15 co-expression modules constructed through WGCNA. Similarly, this approach was evaluated using 10-fold, 100-repeat nested cross-validation. Feature importance analysis identified a relatively uniform distribution of module importance in establishing the predictive model (Figure 6A). The ROC curve for the final combined model achieved a perfect classification performance with an AUC of 1.000 (Figure 6B), demonstrating the strong discriminative capacity. The confusion matrix portrays a similar high classification accuracy, with 14/14 AD cases and 5/7 controls correctly predicted, corresponding to a precision of 0.875 and a recall of 1.000 for the AD class (Figure 6C). Across iterated cross-validations, module-based prediction demonstrates a narrower distribution of AUC, precision, and recall, indicating greater model consistency.

To validate the consistency of our differentially expressed ncRNA findings, we analyzed an independent RNA-sequencing dataset (SRP288246) derived from postmortem brain tissue. Differential expression analysis identified 32 ncRNAs that overlapped between our original dataset and the external validation set (Figure 7). A hypergeometric test yielded a *p*-value of 2.15 × 10^−^⁷, indicating that the observed overlap is highly significant and unlikely to occur by chance. This result supports the robustness and transferability of our findings across independent datasets.

## 4. Discussion

Our comprehensive analysis revealed significant dysregulation of ncRNAs in the plasma of AD patients compared to controls. Specifically, we identified 5164 upregulated and 290 downregulated lncRNAs, 973 upregulated and 26 downregulated piRNAs, and 6 upregulated and 1 downregulated snoRNAs. The relative abundance of each ncRNA class among the differentially expressed genes mirrors their baseline abundance observed during transcript quantification. This proportionality suggests that the representation of each RNA class is not skewed but rather reflects their natural abundance and expression dynamics in plasma.

ncRNAs have been investigated in multiple prior studies as both biomarkers for AD detection and potential drivers of AD. In a systematic review and meta-analysis of circulating lncRNAs as biomarkers for AD diagnosis, the pooled AUC was 0.86, with a total sample size of over 553 AD patients and 513 controls [31]. Our lncRNA panel and combined ncRNA panel achieved significantly higher AUCs of 0.974 and 0.972, respectively. In a more recent study comprising 192 patients, a six-miRNA signature was only able to achieve an AUC of 0.733 [31]. Additionally, our ncRNA panel demonstrates better performance than previously reported combined miRNA-piRNA signature (AUC = 0.83) to distinguish AD and piRNA signature (AUC = 0.86) to predict the conversion of mild cognitive impairment to AD [14]. To the best of our knowledge, snoRNAs have not been used as a diagnostic signature in past studies. The most accurate plasma AD test, a plasma p-tau217 test, is comparable to established CSF assays and reached AUC values between 0.95 and 0.98 [5]. While protein-based biomarkers offer good accuracy, our RNA-based biomarker approach demonstrated similar predictive accuracy. These results reinforce the potential of ncRNA-based biomarkers as a robust, non-invasive diagnostic tool for AD.

Although the top ncRNAs identified in our analysis did not completely overlap with those reported in prior plasma biomarker studies, several of the highly predictive ncRNAs identified in this study have established links to AD-associated molecular pathways. For example, lnc-PRKACB2 is associated with the PRKACB2 protein, which plays a crucial role in tau phosphorylation—a key pathological hallmark of AD [32]. Similarly, lnc-MYEF2-3 is implicated in AD pathogenesis; the knockdown of its associated gene, MYEF2C, has been shown to enhance cellular apoptosis and increase levels of β-secretase (BACE1), a critical enzyme in amyloid-beta production [33]. Among the top-ranked snoRNAs, HBII-180A and HBII-115 are known to guide site-specific 2′-O-methylation of target RNAs, implicating their potential role in RNA modification processes relevant to AD onset [34]. Although HBII-115 (SNORD23) and HBII-180A (SNORD88A) were among the most dysregulated snoRNAs in our dataset, other C/D box snoRNAs, such as SNORD115 and SNORD116, have been reported in plasma extracellular vesicles from individuals with AD [7]. This suggests a potential role for snoRNA in AD-related pathogenesis. Additionally, a statistically significant overlap between differentially expressed ncRNAs in our dataset and those identified in an external validation dataset derived from postmortem brain samples supports the robustness and transferability of our findings across independent cohorts.

To further investigate coordinated ncRNA expression and perform feature selection for AD prediction in a different approach, we conducted a weighted gene co-expression network analysis (WGCNA), identifying 15 distinct co-expression modules. A Random Forest classifier trained on the averaged expression profiles of these modules achieved perfect classification performance (AUC = 1.000) on the test set. The relatively even distribution of feature importance across modules suggests that multiple ncRNA networks contribute collectively to AD prediction. Compared to single-gene and multi-gene models, the module-based classifier also demonstrated a narrower distribution of AUC values across cross-validation iterations, reinforcing the robustness and stability of this module-based predictive approach.

The dysregulated ncRNA landscape is consistent with previous studies, further supporting the involvement of multiple subtypes of ncRNAs in AD pathogenesis and neurodegeneration [35,36]. The substantial dysregulation observed for piRNAs and snoRNAs, despite being less documented in the context of AD, highlights their previously understudied status and suggests the need for further investigation to fully understand their roles in AD onset and progression. Our findings highlight the biological plausibility of our ncRNA panels with strong diagnostic performance, suggesting the importance of further functional characterization of these ncRNA candidates to elucidate their mechanistic roles in AD and assess their translational potential as plasma-based biomarkers.

## 5. Conclusions

This study presents compelling evidence for the diagnostic potential of ncRNA expression profiles, including lncRNA, piRNA, and snoRNA, in Alzheimer’s disease. Through integrative transcriptomic analysis combining differential expression, co-expression networks, and machine learning, we identified robust ncRNA markers capable of distinguishing AD from control samples. However, limitations such as the small sample size and the absence of parallel mRNA-seq data constrain generalizability and mechanistic interpretation. Future directions include validating these findings in larger, independent cohorts and integrating mRNA expression analysis to determine coding-region interaction counterparts of ncRNA biomarkers, thereby enhancing understanding of their functional roles in AD pathogenesis.

## Figures and Tables

**Figure 2 biomolecules-15-00806-f002:**
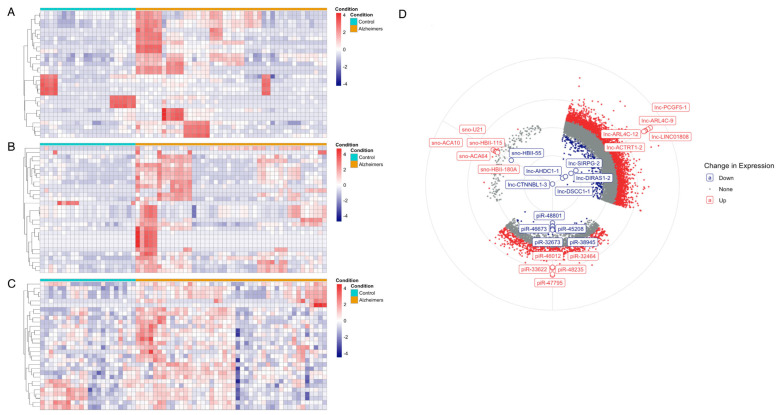
Differential expression of ncRNAs and identification of key features in AD. (**A**–**C**) Heatmaps depicting the top 100 differentially expressed (**A**) lncRNA, (**B**) piRNA, and (**C**) snoRNA across Alzheimer’s (orange) and control (cyan). Normalized expression values (log2-transformed) are expressed on the color scale, where red represents upregulation and blue represents downregulation of expression. (**D**) Circular volcano plot showing the distribution of gene expression across the three classes of ncRNA. Genes with significant upregulation (log2FC ≥ 0.6, padj < 0.05) are shown in red, while significantly downregulated genes are shown in blue. Non-significant genes are plotted in grey. The top 5 most differentially upregulated and top 5 most differentially downregulated genes for each ncRNA class are annotated.

**Figure 3 biomolecules-15-00806-f003:**
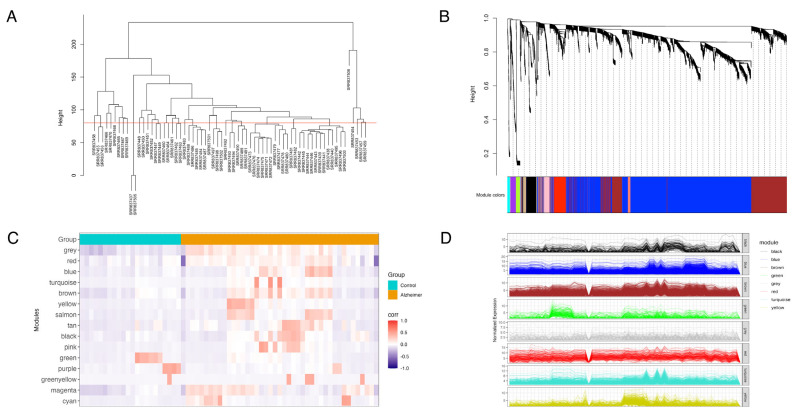
Weighted gene co-expression network analysis (WGCNA) of ncRNAs in AD versus control. (**A**) Sample clustering dendrogram showing hierarchical clustering of samples to detect outliers and assess sample similarity. Samples with similar expression profiles are clustered together based on Euclidean distance. (**B**) Gene clustering dendrogram and module detection illustrate hierarchical clustering of genes based on topological overlap. Module colors are assigned dynamically using the dynamic tree-cut method. Genes with similar expression patterns are grouped into co-expression modules. Modules such as turquoise, blue, and brown contain a high density of correlated genes. (**C**) Module–trait relationship heatmap showing the association between gene modules and sample conditions of Alzheimer’s disease vs. control. The color scale represents correlation coefficients, where red indicates positive correlation and blue indicates negative correlation. The top modules (e.g., turquoise, blue, brown, and red) show distinct correlation patterns with Alzheimer’s disease. (**D**) Module gene expression profiles line plots showing normalized expression of ncRNA in the selected modules (black, blue, brown, green, grey, red, turquoise, and yellow). Individual lines represent the expression profile of an individual gene across samples.

**Figure 4 biomolecules-15-00806-f004:**
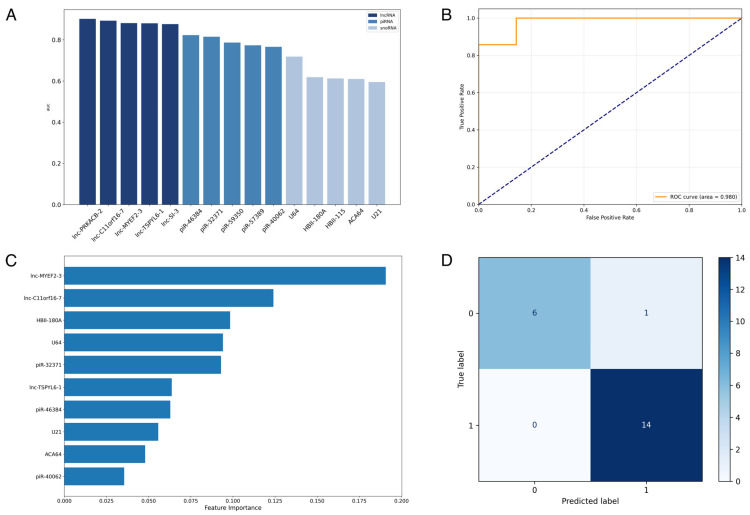
Predictive performance and feature contribution of ncRNA panel for AD diagnosis. (**A**) Area under the curve (AUC) scores of individual non-coding RNA (ncRNA) biomarkers, categorized by RNA type, based on performance in a Random Forest (RF) classification model. (**B**) Receiver-operating characteristic (ROC) curve of the 15-ncRNA panel, demonstrating strong classification performance with an AUC of 0.980. (**C**) Feature importance plot from the trained RF model, highlighting the relative contribution of each ncRNA to predictive accuracy. (**D**) Confusion matrix of model performance on test set, showing high precision (0.933) and perfect recall (1.000) (Label 1 = AD, Label 0 = control).

**Figure 5 biomolecules-15-00806-f005:**
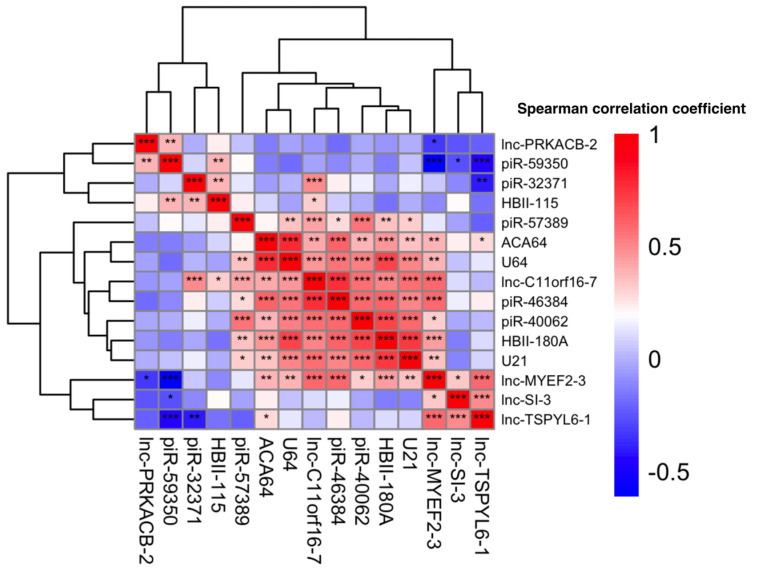
Hierarchical clustering heatmap of pairwise Spearman correlation coefficients among top predictive ncRNAs in Alzheimer’s disease. The heatmap displays correlation strength ranging from −1 to +1. Clustering analysis revealed two major co-expression groups: one cluster including lnc-MYEF2-3, lnc-SI-3, and lnc-TSPYL6-1 with strong positive correlations, and another cluster including lnc-PRKACB-2, piR-59350, and piR-32371 with weaker, more variable correlations. Statistical significance of pairwise correlations is annotated using asterisks: *p* ≤ 0.05 (*), *p* ≤ 0.01 (**), and *p* ≤ 0.001 (***).

**Figure 6 biomolecules-15-00806-f006:**
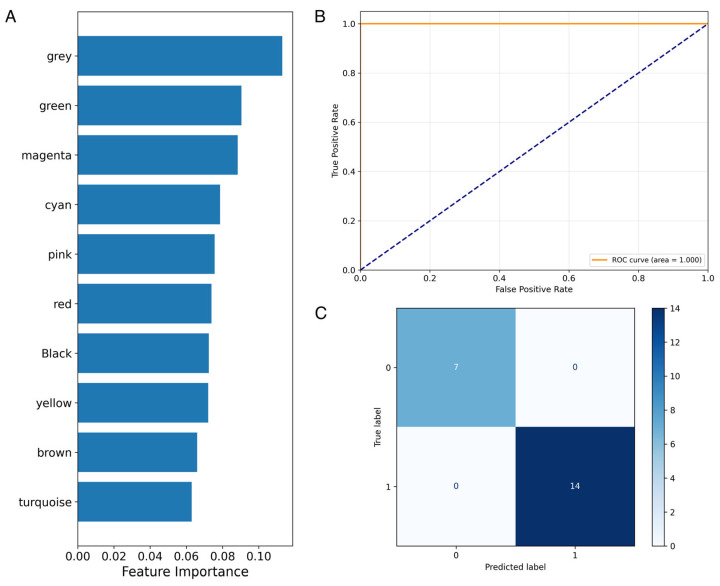
Predictive performance and feature contribution of ncRNA expression modules for Alzheimer’s disease diagnosis. (**A**) Feature importance plot of gene co-expression modules (denoted by module color) from the trained Random Forest (RF) model. (**B**) Receiver operating characteristic (ROC) curve showing perfect classification performance with an AUC of 1.000. (**C**) Confusion matrix of predictions on test set, indicating high precision (0.875) and perfect recall (1.000).

**Figure 7 biomolecules-15-00806-f007:**
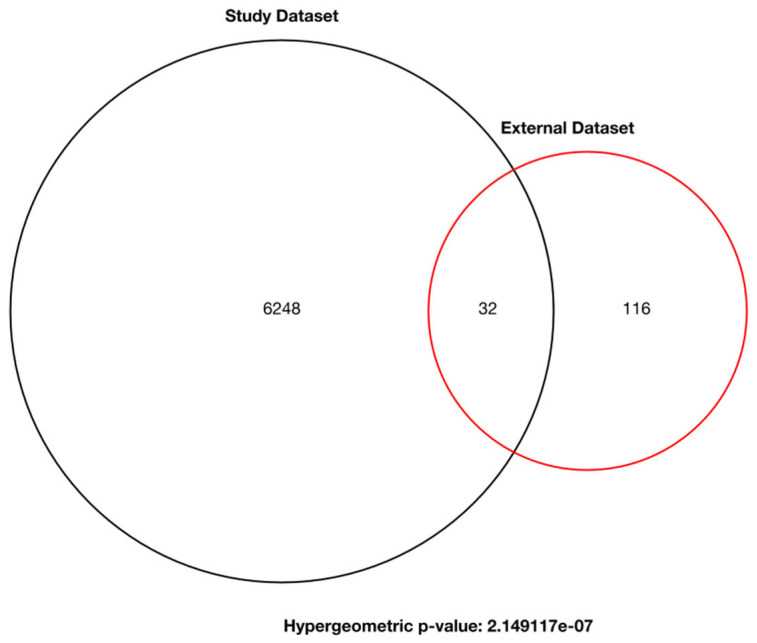
Venn diagram displaying the number of differentially expressed genes common to both the original study dataset and the validation dataset. A total of 32 ncRNAs were found to be commonly expressed in both studies. Statistical testing was performed using the hypergeometric test.

**Table 1 biomolecules-15-00806-t001:** Co-expression modules and relative module size by gene count.

Co-Expression Modules (Color)	Module Size (Gene Count)
Turquoise	2261
Blue	2046
Brown	783
Yellow	425
Green	155
Red	149
Grey	139
Black	136
Pink	71
Magenta	70
Purple	67
Green-yellow	48
Tan	40
Salmon	39
Cyan	30

## Data Availability

The data presented in this study are openly available in the NCBI SRA database under project accession SRP022043 (https://www.ncbi.nlm.nih.gov/Traces/study/?acc=SRP022043&o=acc_s%3Aa) (accessed on 3 March 2025) and project accession SRP288246 (https://www.ncbi.nlm.nih.gov/bioproject/PRJNA670793) (accessed on 20 May 2025).

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
