# Peer review of "Non-Coding RNAs: lncRNA, piRNA, and snoRNA as Robust Plasma Biomarkers of Alzheimer’s Disease"

_biomolecules, 2025, doi:10.3390/biom15060806_

Round 1
Reviewer 1 Report
Comments and Suggestions for Authors
Non-Coding RNAs: lncRNA, piRNA, and snoRNA as Robust 2 Plasma Biomarkers of Alzheimer’s Disease is new and will contribute to the literature on the subject.
My opinion the manuscript is not suited for publication in its present form and the authors should perform some revisions regarding specific points as highlighted herein.
Figures should be more clearly.
The number of samples for the biomarker study is very small, more samples are needed. Also this study needs validation.
Discussion section: Is not well presented and the authors’ efforts to summarize the evidence in this topic of interest is not sufficient. You shoud make revised again
The authors should compare their results with previous studies. What advantages does it provide?
I hope the suggestions for the better of this prepared manuscript will be useful to you.
Author Response
Thank you very much for the constructive criticism. We will address all your critiques point by point.
Comments 1: Figures should be more clearly.
Response 1: Thank you for pointing this out. All figures have been updated to 300 dpi.
Comments 2: The number of samples for the biomarker study is very small, more samples are needed. Also this study needs validation.
Response 2: We agree with your suggestions. Our study is limited by the availability of blood samples sequenced. However, despite the small samples, we show statistical significance in our results. In addition, an additional external validation dataset has been included with 50 Alzheimer’s versus control samples.
Comment 3: Discussion section: Is not well presented and the authors’ efforts to summarize the evidence in this topic of interest is not sufficient. You shoud make revised again
Response 3: The discussion has been revised to emphasize the significance of identifying blood-based biomarkers for Alzheimer’s disease, as they offer a minimally invasive and timely approach to facilitate earlier diagnosis, inform treatment decisions, and potentially improve disease management and patient outcomes.
Comment 4: The authors should compare their results with previous studies. What advantages does it provide?
Response 4: The discussion has been updated to include a comparison between our findings and previous studies, highlighting the strength of our cross-dataset validation and the biological relevance of the identified ncRNAs. Furthermore, our observation of coordinated expression among three distinct classes of ncRNAs underscores the potential for functional interactions and warrants further investigation into their roles in Alzheimer’s disease pathogenesis.
Reviewer 2 Report
Comments and Suggestions for Authors
By evaluating the diagnostic potential of ncRNA expression profiles, this manuscript aims to assess their utility as biomarkers for Alzheimer's disease (AD) classification. The topic is important and timely, particularly for the early diagnosis of AD. I recommend accepting the manuscript after satisfactory revisions addressing the following points:
1. Figure 1 is unclear and difficult to interpret. Please replace it with a higher-resolution version for better readability.
2. Figure 2 is also of low resolution. Please provide a clearer version.
3. The manuscript would benefit from editorial polishing. For consistency and clarity, some terms (e.g., Alzheimer’s disease, non-coding RNA) should be replaced with their commonly used abbreviations (e.g., AD, ncRNA).
Author Response
Thank you very much for the constructive criticism. We will address all your critiques point by point.
Comment 1: Figure 1 is unclear and difficult to interpret. Please replace it with a higher-resolution version for better readability.
Response 1: Thank you for pointing this out. All figures previously submitted and subsequent added figures have been adjusted to have higher resolution.
Comment 2: Figure 2 is also of low resolution. Please provide a clearer version.
Response 2: We aggre with your suggestion. Same as our previous response, all figures previously submitted and newly added figures have been adjusted to have higher resolution.
Comment 3: The manuscript would benefit from editorial polishing. For consistency and clarity, some terms (e.g., Alzheimer’s disease, non-coding RNA) should be replaced with their commonly used
Response 3: Thank you for suggesting. We have adjusted our terminology use throughout the manuscript. In instances of repeatedly used terms such as Alzheimer's Disease, we have used abbreviations and listed them in the abbreviation section of the manuscript for increased clarity.
Reviewer 3 Report
Comments and Suggestions for Authors
The manuscript is interesting; however, some modifications are suggested:
- In "Data Acquisition", on which basis the authors determined the sample size to be totally 70 individuals comprising 48 Alzheimer’s disease patients and 22 healthy controls? why the authors didn't recruit equal number in both groups?
- "Samples SRR837499, SRR837500, and SRR837504 were discarded due to low sample quality", this sentence is ambiguous; did you give your samples any code? please clarify.
- Which type of alignment was used? muscle alignment? clarify.
- I suggest constructing a simple flowchart for the consequences of the study.
- Heading about statistical analyses used should be added to the methodology section.
- All panels in all figures are of poor quality and resolution. The authors must provide better figures. The font inside panels is very small and can't be read.
- Title of the table 1 should be above the table not below.
- I suggest adding a correlation study between the studied biomarkers.
- Discussion part lacks reinforcement of the findings with previously published articles. The authors should add more information and cite more reference to support their conclusions.
- Is the study approved by research ethical committee? if so, add the approval number.
Author Response
Thank you very much for the constructive criticism. We will address all your critiques point by point.
Comment 1: In "Data Acquisition", on which basis the authors determined the sample size to be totally 70 individuals comprising 48 Alzheimer’s disease patients and 22 healthy controls? why the authors didn't recruit equal number in both groups?
Response 1: Thank you for pointing this out. We are using publicly available data and thus are limited with data we have access to. The original study that used this dataset did not explicitly address the reason for the sample imbalance; however, as it was part of a larger project involving multiple neurodegenerative diseases, we surmise that the recruitment strategy was influenced by broader study objectives.
Comment 2:"Samples SRR837499, SRR837500, and SRR837504 were discarded due to low sample quality", this sentence is ambiguous; did you give your samples any code? please clarify.
Response 2: We agree with your suggestion. We have clarified that during quality control procedures using FastQC, we identified samples SRR837499, SRR837500, and SRR837504 as having particularly low Per Sequence Quality Scores, indicating poor overall read quality. Including these samples would have resulted in substantial read loss during trimming and compromised downstream analyses. Therefore, they were excluded from subsequent processing to ensure the integrity and consistency of the dataset. (line 112)
Comment 3: Which type of alignment was used? muscle alignment? clarify.
Response 3: Thank you for pointing this out. We did not perform multiple sequence alignment such as MUSCLE in this study. Instead, we aligned to mapping RNA-seq reads to the human reference genome (hg38) using the STAR aligner (v2.7.2b). STAR performs spliced alignment optimized for RNA-seq data, allowing for accurate mapping of reads for ncRNA. (line 116)
Comment 4: I suggest constructing a simple flowchart for the consequences of the study.
Response 4: We agree wth your suggestion. A flow chart has been created (Fig1, line 145) to address the overall workflow of the study including external validation.
Comment 5: Heading about statistical analyses used should be added to the methodology section.
Response 5: Thank you for suggesting. We have included a short paragraph on the statistical analysis used in the methodology section. (line 139)
Comment 6: All panels in all figures are of poor quality and resolution. The authors must provide better figures. The font inside panels is very small and can't be read.
Response 6: We agree with your suggestion. All previous figures and newly added figures have been adjusted to have higher resolution with font sizes adjusted.
Comment 7: Title of the table 1 should be above the table not below.
Response 7: Thank you for pointing out. The title has been moved above the table.
Comment 8: I suggest adding a correlation study between the studied biomarkers.
Response 8: This is an excellent suggestion, and accordingly, an additional result section has been added that involves pair-wise correlation study which suggests strong expression patterns between the top predictive biomarkers (line 259).
Comment 9: Discussion part lacks reinforcement of the findings with previously published articles. The authors should add more information and cite more reference to support their conclusions.
Response 9: Thank you for pointing this out. The discussion has been updated to include a comparison between our findings and previous studies, highlighting the strength of our cross-dataset validation and the biological relevance of the identified ncRNAs. Furthermore, our observation of coordinated expression among three distinct classes of ncRNAs underscores the potential for functional interactions and warrants further investigation into their roles in Alzheimer’s disease pathogenesis. Additional references have also been added to support our conclusion. (line 299)
Comment 10: Is the study approved by research ethical committee? if so, add the approval number.
Response 10: Thank you for your suggestion. We used publicly available de-identified datasets in which we extracted gene expression from the sequencing data. Thus, we do not require an IRB for this. In addition, the original dataset states adherence to ethical, confidentiality, and social and legal issues on the use of human subjects.
This is appended below:
Original dataset:
Samples and clinical data supplied by PrecisionMed are handled in strictest compliance with all applicable rules and regulations including the recommendations of the Council of the Human Genome Organization (HUGO) Ethical, Legal, and Social Issues Committee (HUGO-ELSI, 1998); with the United Nations Educational, Scientific, and Cultural Organization's (UNESCO) Universal Declaration on the Human Genome and Human Rights (1997); and with recommendations guiding physicians in biomedical research involving human subjects adopted by the 18th World Medical Assembly, Helsinki, Finland, 1964 and later revisions.
Validation dataset:
The studies involving human participants were reviewed and approved by WIRB-Copernicus Group, Inc. The patients/participants provided their written informed consent to participate in this study.
Round 2
Reviewer 1 Report
Comments and Suggestions for Authors
The necessary arrangements have been made and are suitable for publication.
Reviewer 3 Report
Comments and Suggestions for Authors
Thank you for your effort. All comments have been addressed.